# Exposure to air pollution and scarlet fever resurgence in China: a six-year surveillance study

Yonghong Liu[1,11], Hui Ding[1,11], Shu-ting Chang[2,11], Ran Lu[3], Hui Zhong[1], Na Zhao [4], Tzu-Hsuan Lin[2], Yiming Bao [5], Liwei Yap[2], Weijia Xu[6], Minyi Wang[6], Yuan Li[7], Shuwen Qin [8], Yu Zhao [8], Xingyi Geng [9], Supen Wang [10], Enfu Chen [8✉], Zhi Yu[1✉], Ta-Chien Chan [2✉] & Shelan Liu[8✉]

Scarlet fever has resurged in China starting in 2011, and the environment is one of the potential reasons. Nationwide data on 655,039 scarlet fever cases and six air pollutants were retrieved. Exposure risks were evaluated by multivariate distributed lag nonlinear models and a meta-regression model. We show that the average incidence in 2011–2018 was twice that in 2004–2010 [RR = 2.30 (4.40 vs. 1.91), 95% CI: 2.29–2.31; p < 0.001] and generally lower in the summer and winter holiday (p = 0.005). A low to moderate correlation was seen between scarlet fever and monthly $NO_2$ (r = 0.21) and $O_3$ (r = 0.11). A 10 µg/m$^3$ increase of $NO_2$ and $O_3$ was significantly associated with scarlet fever, with a cumulative RR of 1.06 (95% CI: 1.02–1.10) and 1.04 (95% CI: 1.01–1.07), respectively, at a lag of 0 to 15 months. In conclusion, long-term exposure to ambient $NO_2$ and $O_3$ may be associated with an increased risk of scarlet fever incidence, but direct causality is not established.

[1] School of Intelligent Systems Engineering, Sun Yat-sen University, Guangzhou, Guangdong Province, China. [2] Research Center for Humanities and Social Sciences, Academia Sinica, Taipei, Taiwan. [3] Division of Infectious Disease, Key Laboratory of Surveillance and Early-Warning on Infectious Disease, Chinese Center for Disease Control and Prevention, Beijing, China. [4] CAS Key Laboratory of Pathogenic Microbiology and Immunology, Institute of Microbiology, Chinese Academy of Sciences, Beijing, China. [5] National Genomics Data Center, Beijing Institute of Genomics, Chinese Academy of Sciences & China National Center for Bioinformation, Beijing, China. [6] Guangdong Provincial Key Laboratory of Intelligent Transport System, Guangzhou, Guangdong Province, China. [7] Department of Infectious Diseases, Baoan District Centre for Disease Control and Prevention, Shenzhen, Guangdong Province, China. [8] Department of Infectious Diseases, Zhejiang Provincial Centre for Disease Control and Prevention, Hangzhou, Zhejiang Province, China. [9] Emergency Offices, Jinan Centre for Disease Control and Prevention, Jinan, Shandong Province, China. [10] College of Life Sciences, Anhui Normal University, Wuhu, Anhui Province, China. [11] These authors contributed equally: Yonghong Liu, Hui Ding, Shu-ting Chang. ✉email: enfchen@cdc.zj.cn; stsyuz@mail.sysu.edu.cn; dachianpig@gmail.com; liushelan@126.com

Scarlet fever is a disease caused by a group A Streptococcus (*Streptococcus pyogenes*)[1,2]. The signs and symptoms include a sore throat, fever, headaches, swollen lymph nodes, and a characteristic rash[3]. It most commonly affects children between 5 and 15 years of age, though it can also occur in adults[4].

Most scarlet fever cases are mild infections, and fatal infections are now rare[5]. However, a small proportion of cases still may develop serious sequelae, including kidney disease, rheumatic heart disease, and arthritis[2,6]. Spread of scarlet fever occurs by close contact, via respiratory droplets (e.g., saliva or nasal discharge) or by fomites[4]. Scarlet fever outbreaks are most often observed in kindergartens, other schools, factories, etc.[7–10].

The incidence of most childhood infectious diseases significantly decreased during the course of the last century with immunization, development of effective treatments, improved living standards, hygiene, nutrition, etc.[1,2]. Scarlet fever was a feared killer in the eighteenth and nineteenth centuries throughout Europe. Its spread dropped dramatically worldwide during the twentieth century[11], but it has recently reemerged, as evidenced by its rise in several countries[8,12]. For example, increased scarlet fever incidence has been seen in Asia, especially in Korea, mainland China and Hong Kong, and in Europe, particularly in the United Kingdom[1,5,12–15]. Because there are currently no vaccines available to protect against *S. pyogenes* infection, the resurgence of scarlet fever has been a concerning public health problem globally[8].

In 2019, the World Health Organization (WHO) listed ten threats to global health, of which air pollution is considered by the WHO to be the greatest environmental risk to health. The primary cause of air pollution is also a major contributor to climate change, which impacts people's health in different ways. A string of evidence has revealed positive associations between air pollution exposure and respiratory diseases[16]. However, the effects of air pollution on resurgence of scarlet fever have been less reported in both developed and developing countries. A small number of previous studies have projected air-pollution-related scarlet fever in a small number of cases, but these only focused on specific regions or cities in China, or were based on a single scenario[12,17–19], and the findings are diverse, fragmental, and nonconclusive.

In this study, we applied an ecological study design to examine the associations between long-term air pollution exposure, meteorological conditions, and scarlet fever incidence in all of China. We assessed the relative risk by a distributed lag nonlinear model (DLNM), stratified by different air pollutants, meteorological factors, and high- and low-incidence areas. In addition, the demographic and behavioral effects on scarlet fever incidence such as population density and school breaks were also evaluated. To our knowledge, this is the first nationwide study of the relationship between historical exposure to air pollution exposure and a sudden rise in scarlet fever, relying on the largest data from all parts of China, covering an overall population and the longest period.

These findings showed that the number of scarlet fever cases began to increase suddenly from 2011. Statistical examinations of 6-year nationwide data suggested long-term exposure to ambient $NO_2$ and $O_3$ is associated with the scarlet fever upsurge. Scarlet fever incidence also appeared to be associated with school breaks such that lower incidence rates were observed in the summer and winter holidays compared with when school was in session. Despite the inherent limitations of the ecological study design, this study encourages public health authorities to consider $NO_2$ and $O_3$ risks when addressing the prevention and control of scarlet fever resurgence. School-based control measures could be particularly important in scarlet fever control.

## Results

**Scarlet fever distribution in China, 2004–2018.** The study consisted of 655,039 scarlet fever cases between January 1, 2004 and December 31, 2018. The annualized average incidence was 3.26 per 100,000 people (shown in Fig. 1). Scarlet fever started to surge in 2011, rising by a factor of three between 2004 and 2011 (rate ratio [RR] 3.27, 95% CI: 3.22–3.32; $p < 0.001$), further increasing in 2017 (5.37 per 100,000 population) and peaking in 2018 (5.67 per 100,000 population) (shown in Fig. 1). The average incidence rate during the post-upsurge period (2011–2018) was two times more than in the pre-upsurge period (2004–2010) [RR = 2.30 (4.40 vs. 1.91), 95% CI: 2.29–2.31; $p < 0.001$] (see Fig. 1).

We used a heat map to show seasonal patterns, and found that nationally, scarlet fever showed semiannual peaks of activity, including a major peak in May and June followed by a smaller peak in November and December (Supplementary Fig. 1). Scarlet fever predominantly circulated in the north, northeast, and northwest of China (Supplementary Fig. 2).

**Air pollution characteristics in China, 2013–2018.** During 2013–2018, nationally, the monthly mean concentration was 51.28 μg/m³ for $PM_{2.5}$, 90.75 μg/m³ for $PM_{10}$, 24.35 μg/m³ for $SO_2$, 33.63 μg/m³ for $NO_2$, and 1.08 mg/m³ for CO, and the daytime 8-h mean concentration for $O_3$ was 86 mg/m³. The monthly concentrations of $PM_{2.5}$ and $PM_{10}$ were much higher than the China guidelines II level issued in 2018 (see Table 1 and Supplementary Fig. 3). The boxplots of monthly variation of air pollution concentrations show an obvious seasonal pattern (Fig. 2a). The peaks of $PM_{2.5}$, $PM_{10}$, and $NO_2$ concentration mostly appeared in December and January, while the peaks of $O_3$ appeared in late spring to late summer, from May to August.

The annual mean concentrations of six air pollutants varied greatly across the 31 provinces, but significantly increased in northern to western China (Fig. 3 and Supplementary Fig. 3). In particular, the concentrations of $PM_{2.5}$ and $PM_{10}$ in the majority of provinces were over the China guidelines II (Fig. 3). During 2013–2018, the mean monthly concentrations of six air pollutants in high-latitude areas were obviously higher than those in low-latitude areas (Supplementary Fig. 4 and Supplementary Table 1). Nevertheless, their trends varied over time. The mean monthly concentrations of $PM_{2.5}$, $PM_{10}$, and CO significantly decreased year by year, while the values of $O_3$ greatly increased in the 6 years during 2013–2018, and $NO_2$ showed a volatile rising trend starting in 2016, following a downward trend during 2013–2016. The concentration of $PM_{2.5}$, $PM_{10}$, and $NO_2$ in most months exceeded the China guidelines II. Meanwhile, the seasonal variation in the high-latitude areas was basically consistent with that in the low-latitude areas (Supplementary Fig. 4). In addition, the concentrations of $NO_2$ and $O_3$ were positively correlated with scarlet fever incidences in quantile groups (Supplementary Fig. 5).

**Meteorological factors distribution in China, 2004–2018.** Nationally, the monthly mean ambient temperature was 13.37 °C, the mean relative humidity was 66.03%, air pressure was 940.27 Pa, precipitation amount was 76.90 mm, wind speed was 2.13 m/s, and sunlight was 172.59 h throughout China. The values after imputation are listed in brackets in Table 2, and are very close to the original data distribution. We adopted the imputed values in our models.

There was significantly higher precipitation than that just in the period of 2004–2010, whereas the hours of sunlight and wind speeds were much lower in the post- than pre-upsurge period (all $p$ values < 0.05, Supplementary Fig. 6). In contrast, mean temperature, relative humidity, and air pressure were not

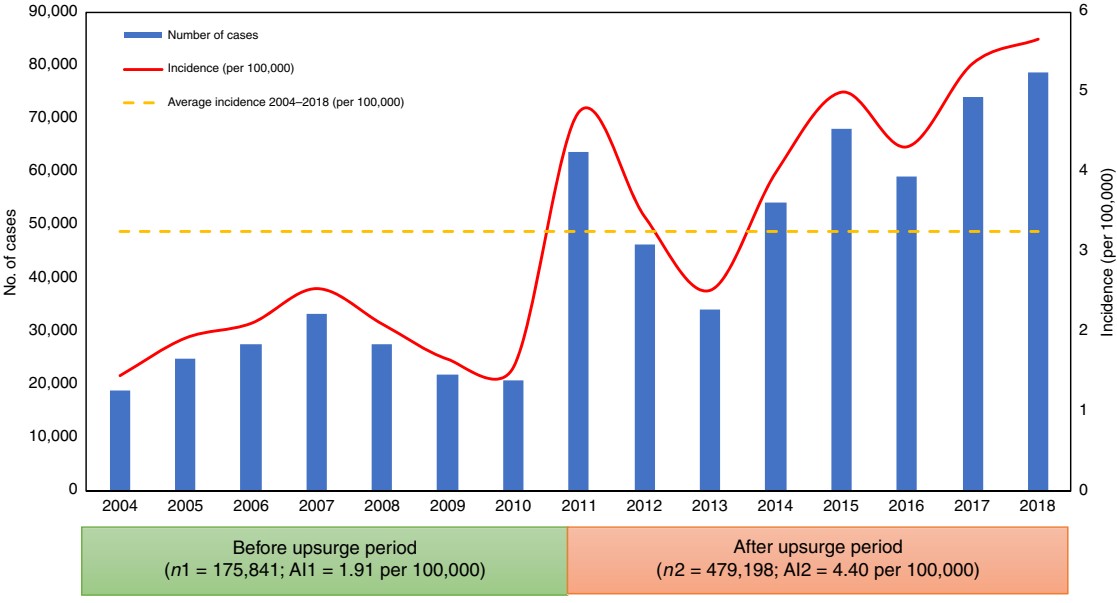

**Fig. 1 Annual incidence rate of scarlet fever in all of China, 2004–2018.** Blue bar: the number of scarlet fever notifications by year; Red line: Annual scarlet fever incidence per 100,000 population by calendar year; Yellow line: the mean annual incidence for the period 2004–2018; We defined the mean annual incidence as the cumulative number of annual scarlet fever cases (2004–2018) divided by the general population size (2004–2018); AI1 = average annual incidence 2004–2010 (per 100,000). We defined AI1 as the cumulative number of annual scarlet fever cases (2004–2010) divided by the general population size (2004–2010); AI2 = average annual incidence 2011–2018 (per 100,000). We defined AI2 as the cumulative number of annual scarlet fever cases (2011–2018) divided by the general population size (2011–2018). The figure consists of 655,039 scarlet fever cases between January 1, 2004 and December 31, 2018. Scarlet fever started to surge in 2011 and peaked in 2018 (5.67 per 100,000 population). After the upsurge in 2011, the annual incidence of scarlet fever remained higher than average level except for in 2013. Source data are provided as a Source Data file.

**Table 1 Descriptive statistics for monthly scarlet fever cases and air pollution concentrations, and weather conditions in China, 2013–2018 ($n = 2232$).**

| Variables | Mean | SD | Min. | $P_{25}$ | $P_{50}$ | $P_{75}$ | Max. | §China guideline II |
|---|---|---|---|---|---|---|---|---|
| No. of scarlet fever cases | 5123 | 2893 | 959 | 2704 | 4445 | 7262 | 12593 | / |
| $PM_{2.5}$ (µg/m³) | 51.28 | 29.09 | 8.00 | 31.00 | 44.00 | 64.00 | 225.00 | 35 |
| $PM_{10}$ (µg/m³) | 90.75 | 44.56 | 20.00 | 58.00 | 82.00 | 115.00 | 405.00 | 70 |
| $SO_2$ (µg/m³) | 24.35 | 22.00 | 2.00 | 12.00 | 17.00 | 28.25 | 228.00 | 60 |
| $NO_2$ (µg/m³) | 33.63 | 13.82 | 9.00 | 23.00 | 32.00 | 42.00 | 97.00 | 40 |
| $O_3$ (µg/m³) | 86.00 | 31.52 | 9.00 | 63.00 | 84.00 | 108.00 | 198.00 | 160 |
| CO (mg/m³) | 1.08 | 0.47 | 0.30 | 0.80 | 1.00 | 1.20 | 4.90 | 4 |
| Mean temperature (°C) | 13.54 | 10.78 | −22.68 | 6.49 | 14.91 | 22.21 | 32.00 | / |
| Relative humidity (%) | 66.71 | 13.72 | 27.23 | 56.74 | 69.63 | 78.33 | 89.21 | / |
| Air pressure (Pa) | 939.98 | 94.10 | 644.76 | 904.73 | 984.44 | 1001.48 | 1030.95 | / |
| Precipitation (mm)[a] | 74.50 (74.53) | 72.66 (72.43) | 0.00 (0.00) | 14.22 (14.58) | 52.28 (52.42) | 118.12 (118.20) | 570.30 (570.30) | / |
| Wind speed (m/s) | 2.17 | 0.50 | 1.07 | 1.80 | 2.12 | 2.49 | 4.30 | / |
| Sunlight (h)[a] | 170.67 (170.50) | 59.24 (58.76) | 17.80 (17.80) | 129.59 (129.93) | 176.30 (175.45) | 214.21 (213.85) | 328.57 (328.57) | / |

/: Not available.
(.) values after imputation, *SD* standard deviation, *min.* minimum, $P_{25}$ 25th percentile, $P_{50}$ median, $P_{75}$ 75th percentile, *max.* maximum, $PM_{2.5}$ particulate matter of <2.5 µm, $PM_{10}$ particulate matter of <10 µm, $SO_2$ sulfur dioxide, $NO_2$ nitrogen dioxide, $O_3$ ozone, *CO* carbon monoxide.
[a]There were a number of missing values in the following variables: precipitation: 25, sunlight: 61.

significantly different between the two periods (all *p* values > 0.05, Supplementary Fig. 6).

The boxplots of meteorological conditions show clear variations in the four seasons (Fig. 2b) from 2013 to 2018, and the incidence of scarlet fever also showed seasonal variations, with the average highest number of cases in spring (Fig. 2c), and a similar pattern can also be found from 2004 to 2018 (Supplementary Fig. 7). The

temperature, relative humidity, and precipitation were higher in summer. The atmospheric pressure was higher in winter, and wind speed and sunlight were higher in spring.

We further stratified the analyses of meteorological variables at higher latitudes and low latitudes, finding that higher latitude areas showed lower mean temperature, lower relative humidity, lower pressure, and lower precipitation amount. However, higher

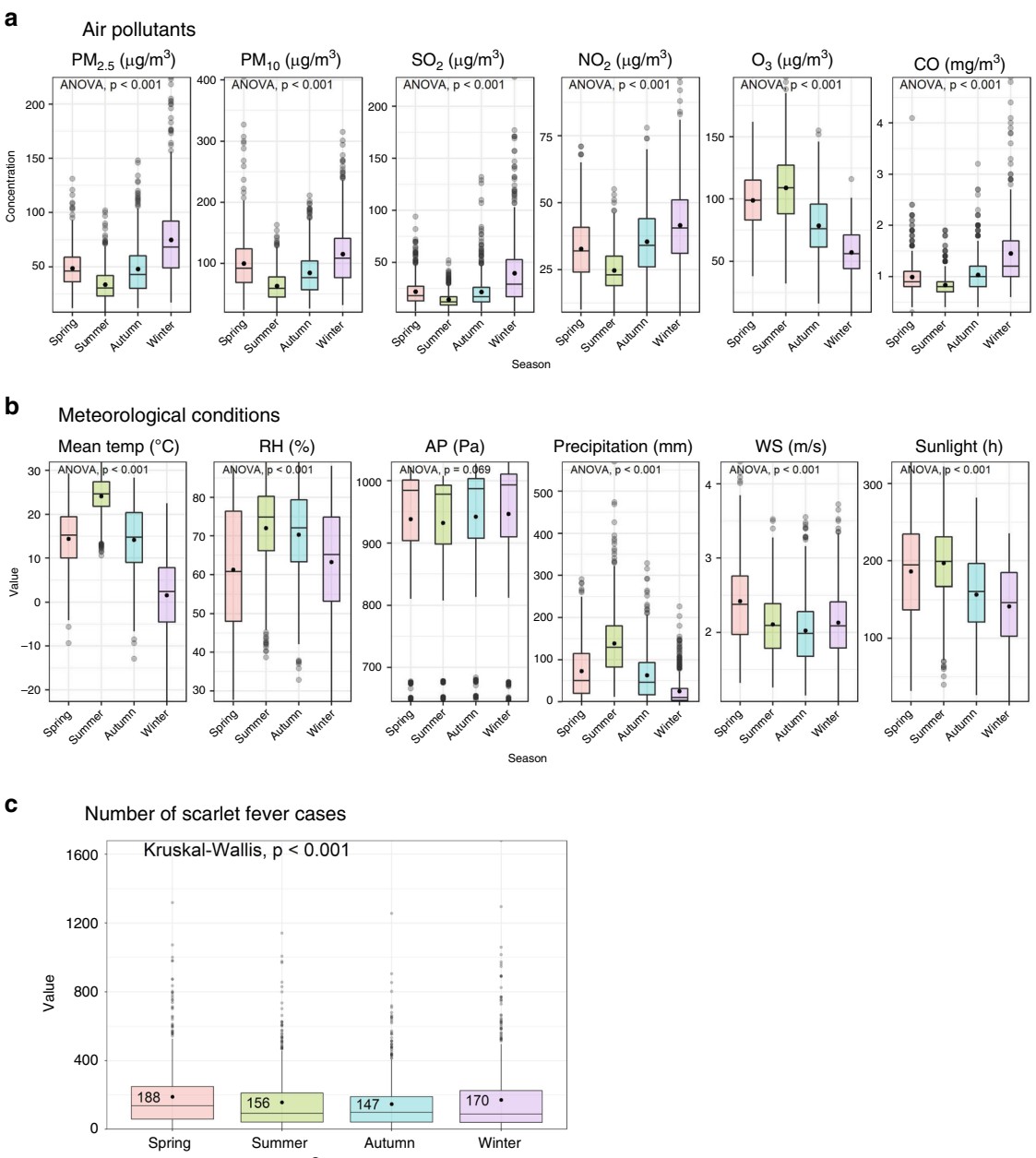

**Fig. 2 Boxplots of six air pollutants, six meteorological conditions and the number of scarlet fever cases in four seasons from 2013 to 2018. a** The seasonal pattern of air pollutants. **b** The seasonal pattern of weather conditions. **c** The seasonal pattern of scarlet fever. An analysis of variance (ANOVA) test is applied to examine the values or concentrations among the four seasons. A Kruskal–Wallis test is used to examine the scarlet fever incidences among the four seasons: spring (March to May), summer (June to August), autumn (September to November), and winter (December to February). $PM_{2.5}$ particulate matter with aerodynamic diameter <2.5μm. $PM_{10}$ particulate matter with aerodynamic diameter <10μm, $SO_2$ sulfur dioxide, $NO_2$ nitrogen dioxide, $O_3$ ozone, CO carbon monoxide, Mean Temp mean temperature, RH relative humidity, AP air pressure, WS wind speed. Source data are provided as a Source Data file.

wind speed and sunshine were identified in high-latitude areas (all $p$ values < 0.05; see Supplementary Fig. 8 and Supplementary Table 2).

**Relationship between air pollutants and scarlet fever.** We discovered significant associations between scarlet fever incidence and four of the six air pollutants. The strongest correlation was found in $NO_2$ [$r = 0.21$], while the other positive correlations were weak [$PM_{10}$ ($r = 0.13$), max 8-h average of ozone ($r = 0.11$), and $PM_{2.5}$ ($r = 0.06$)] (see Fig. 4).

The exposure–response relationship curves showed essentially linear associations between scarlet fever and lag air pollution concentrations in the DLNM model. In the single-variable model (Fig. 5a–d), the range of relative risks from 2013 to 2018 was 0.54–2.04 for $NO_2$, 0.65–1.92 for $O_3$, 0.39–3.24 for $PM_{10}$, and 0.35–1.78 for $PM_{2.5}$. The remaining two air pollutants, $SO_2$ and CO, were not significantly correlated with scarlet fever incidence ($p > 0.05$). Thus, we did not include them in DLNM. In the multiple-variable model (Fig. 6a, b), the range of relative risks from 2013 to 2018 was 0.73–2.44 for $NO_2$ and 0.91–1.14 for $O_3$. The maximum RR of $NO_2$ was 2.44 under exposure to 97 μg/m³

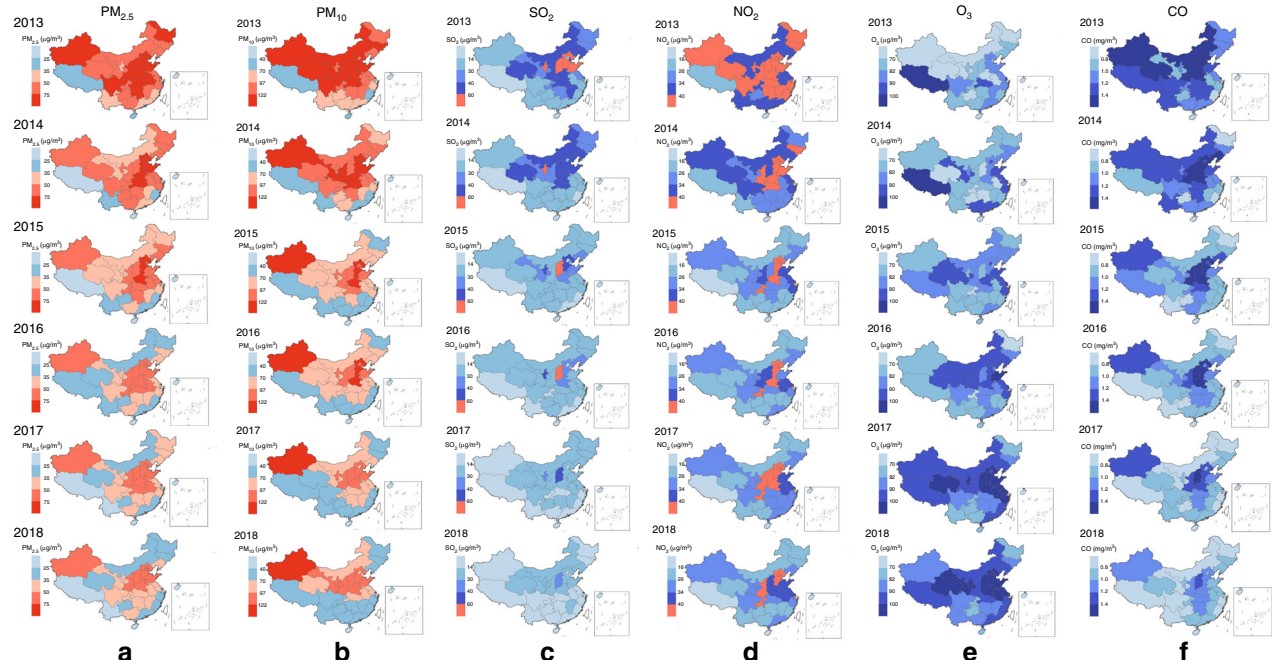

**Fig. 3 Changes of spatiotemporal distribution of six air pollutants in China, 2013–2018. a** The changes of average annual value of particulate matter of <2.5 μm ($PM_{2.5}$) during 2013–2018 in 31 provinces of China. **b** The changes of average annual value of particulate matter of <10 μm ($PM_{10}$) during 2013–2018 in 31 provinces of China. **c** The changes of average annual value of sulfur dioxide ($SO_2$) during 2013–2018 in 31 provinces of China. **d** The changes of average annual value of nitrogen dioxide ($NO_2$) during 2013–2018 in 31 provinces of China. **e** The changes of average annual value of ozone ($O_3$) during 2013–2018 in 31 provinces of China. **f** The changes of average annual value of carbon monoxide (CO) during 2013–2018 in 31 provinces of China. Choropleth maps of average annual value of air pollutants by region. Red means over China guidelines II (issued 2018); Blue: means below China guidelines II. Depth of color denotes air pollutants concentration. Source data are provided as a Source Data file deposited on a publicly available website (https://doi.org/10.6084/m9.figshare.12237596.v2).

---

**Table 2 Descriptive statistics for monthly scarlet fever cases and weather conditions in China, 2004–2018 ($n = 5580$).**

| Variables | Mean | SD | Min. | $P_{25}$ | $P_{50}$ | $P_{75}$ | Max. |
|---|---|---|---|---|---|---|---|
| No. of scarlet fever cases | 3642 | 2618 | 423 | 1766 | 2860 | 4652 | 12593 |
| Mean temperature (°C)[a] | 13.37 (13.36) | 10.87 (10.87) | −23.12 (−23.12) | 6.16 (6.16) | 14.88 (14.88) | 22.12 (22.11) | 32.00 (32.00) |
| Relative humidity (%)[a] | 66.03 (66.03) | 13.03 (13.04) | 27.23 (27.23) | 57.53 (57.52) | 68.83 (68.84) | 76.53 (76.53) | 89.45 (89.45) |
| Air pressure (Pa)[a] | 940.27 | 94.10 | 644.76 | 904.86 | 984.96 | 1002.00 | 1032.20 |
| Precipitation (mm)[a] | 76.90 (76.87) | 81.80 (81.68) | 0.00 (0.00) | 14.94 (14.98) | 51.98 (51.98) | 114.00 (114.41) | 1055.26 (1055.26) |
| Wind speed (m/s)[a] | 2.13 (2.13) | 0.50 (0.50) | 0.84 (0.84) | 1.78 (1.78) | 2.09 (2.09) | 2.44 (2.44) | 4.10 (4.10) |
| Sunlight (hours)[a] | 172.59 (172.44) | 58.84 (58.70) | 11.90 (11.90) | 131.95 (131.96) | 176.68 (176.51) | 215.63 (215.33) | 328.57 (328.57) |

(.) Values after imputation, *SD* standard deviation, *min.* minimum, $P_{25}$ 25th percentile, $P_{50}$ median, $P_{75}$ 75th percentile, *max.* maximum.
[a]There were a number of missing values in the following variables: mean temperature: 3, relative humidity: 3, air pressure: 3, precipitation: 26, wind speed: 3, and sunlight: 63.

---

of $NO_2$ at lag 0 months. The maximum RR of $O_3$ was 1.14 under exposure to 198 of $O_3$ at lag 0 months. $PM_{2.5}$ and $PM_{10}$ were not included in the multiple-variables model because of their collinear relationship with $NO_2$.

**Relationship between weather condition and scarlet fever.** The DLNM model further discovered that five of the six meteorological variables (all but air pressure) had a significant association with scarlet fever incidence. Two meteorological conditions showed positive relationships with the disease: monthly sunlight ($r = 0.27$) and wind speed ($r = 0.24$); by contrast, it was inversely correlated with monthly relative humidity (RH, $r = -0.37$),

precipitation ($r = -0.25$), and mean temperature ($r = -0.2$); see Fig. 4.

The exposure–response relationship curves showed essentially linear associations between scarlet fever and lag meteorological conditions in the DLNM model. In the single-variable model (Fig. 5e–i), the range of relative risks from 2004 to 2018 was 0.66–2.05 for sunlight, 0.63–1.84 for wind speed, 0.63–2.31 for relative humidity, 0.00–6.81 for precipitation, and 0.08–5.40 for mean temperature. In the multiple-variables model (Fig. 6c–g), the range of relative risks from 2013 to 2018 was 0.68–1.31 for sunlight, 0.76–1.29 for wind speed, 0.82–1.32 for relative humidity, 0.02–1.26 for precipitation, and 0.20–2.64 for mean temperature. The maximum RR of sunlight is 1.31 under 328 h at lag 0 months.

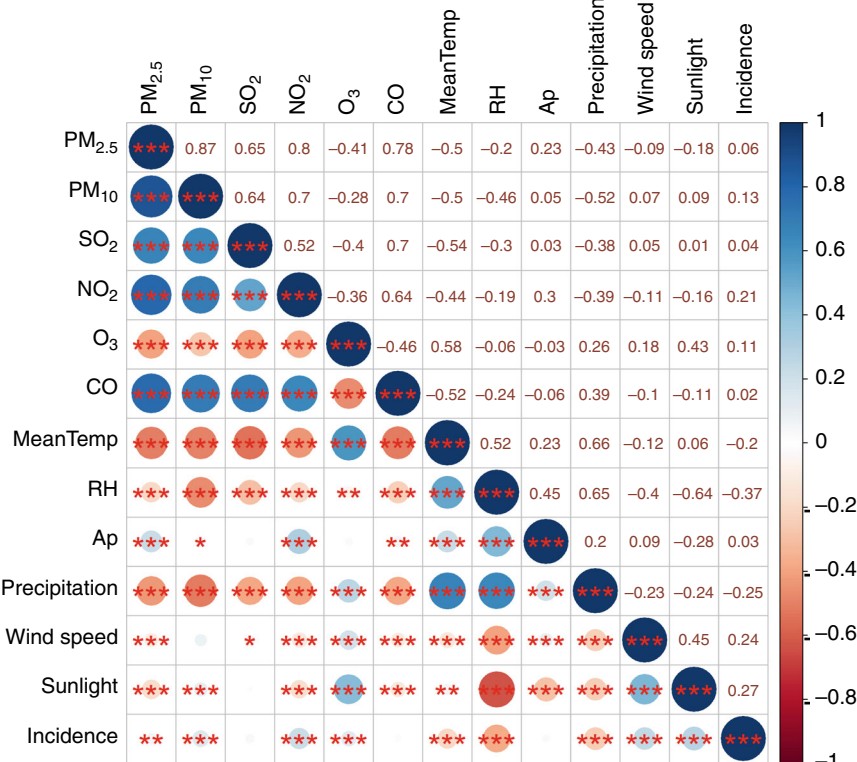

**Fig. 4 Pearson correlation coefficients between air pollution concentrations and weather conditions and scarlet fever incidence in China, 2013–2018 ($n = 2232$).** $PM_{2.5}$ particulate matter with aerodynamic diameter <2.5 μm, $PM_{10}$ particulate matter with aerodynamic diameter <10 μm, $SO_2$ sulfur dioxide, $NO_2$ nitrogen dioxide, $O_3$ ozone, CO carbon monoxide, MeanTemp mean temperature, RH relative humidity, AP air pressure. *means $0.05 \geq p > 0.01$; **means $0.01 \geq p > 0.001$; ***means $\leq 0.001$. Source data are provided as a Source Data file.

The maximum RR of wind speed is 1.29 under 4 m/s at lag 0.8 months. The maximum RR of relative humidity is 1.32 under 28% at lag 1.2 months. The maximum RR of precipitation is 1.26 under 398 mm at lag 0 months. The maximum RR of mean temperature is 2.64 under −22 °C at lag 15 months.

**Cumulative risks at lag 0–15 months**. We found that the cumulative risks at lag 0–15 months of long-term exposure to ambient $NO_2$ were associated with scarlet fever incidence, and also found that high $O_3$, low mean temperature, and low wind speed were associated with an increase in scarlet fever incidence (Fig. 7). Nonlinear associations were found among sunlight hours, relative humidity, and precipitation. In addition, $NO_2$ and $O_3$ are significantly associated with scarlet fever incidence ($NO_2$: with reference to 40 μg/m³; $O_3$: with reference to 160 μg/m³) with a cumulative RR of 1.06 (95% CI: 1.02–1.10) and 1.04 (95% CI: 1.01–1.07), respectively, at a lag of 0−15 months by the multiple-variables model (Supplementary Table 3).

**School breaks, demographic effect, and surging effect**. We analyzed the associations between scarlet fever incidence and behavioral factors, including school closures during the summer and winter breaks, and population density in each province. First, we computed the two kinds of average scarlet fever incidence, the first during summer break (July and August) and winter break (January and February), and the second for the remaining months in each province. In Fig. 8, the innermost ring represents the average monthly incidence when school is in session, and the outermost ring represents the average monthly incidence during school summer and winter breaks. It is clearly shown that the incidence is generally higher in the remaining months, and the independent $t$ test also shows that the mean difference in the

remaining months is statistically significantly higher than those during breaks (mean difference: 0.295 per 100,000 population, $p = 0.005$).

Second, we applied DLNM and meta-regression to elucidate how high and low population density is associated with the risk of scarlet fever incidence from 2013 to 2018 (Supplementary Fig. 9a). The predicted curves from meta-regression for the 25th (red line, population density = 135.7 persons/km²) and 75th (green line, population density = 480.4 persons/km²) percentiles of population show there are no significant differences of $NO_2$ risks referenced at the 15th percentile of $NO_2$ concentration (23.32 μg/m³). In addition, the predicted curves for the 25th (red line, incidence = 0.2/100,000) and 75th (green line, incidence = 0.7/100,000) percentiles of incidence show there are no significant differences of $PM_{2.5}$ and $PM_{10}$ risks referenced at the 15th percentile of $PM_{2.5}$ concentration (30.49 μg/m³) and $PM_{10}$ concentration (58.58 μg/m³), respectively (Supplementary Fig. 9b, c). Third, the predicted exposure–response relationship in terms of relative risk between precipitation, wind speed, and sunlight had no significant difference before and after 2011 (Supplementary Fig. 10a–c).

## Discussion

Using a 15-year national infectious diseases dataset, meteorological surveillance, and a 6-year air quality surveillance database, we found that China experienced an unexplained resurgence of scarlet fever in 2011, which continued and then peaked in 2018. The monthly incidence was generally lower in the summer and winter holiday throughout a year. This study revealed that among the various air pollutants examined, the risk estimates for $NO_2$ and $O_3$ were the most robust in the DLNM model. The concentrations of $NO_2$ and $O_3$, traffic-related pollutants, had low to moderate positive

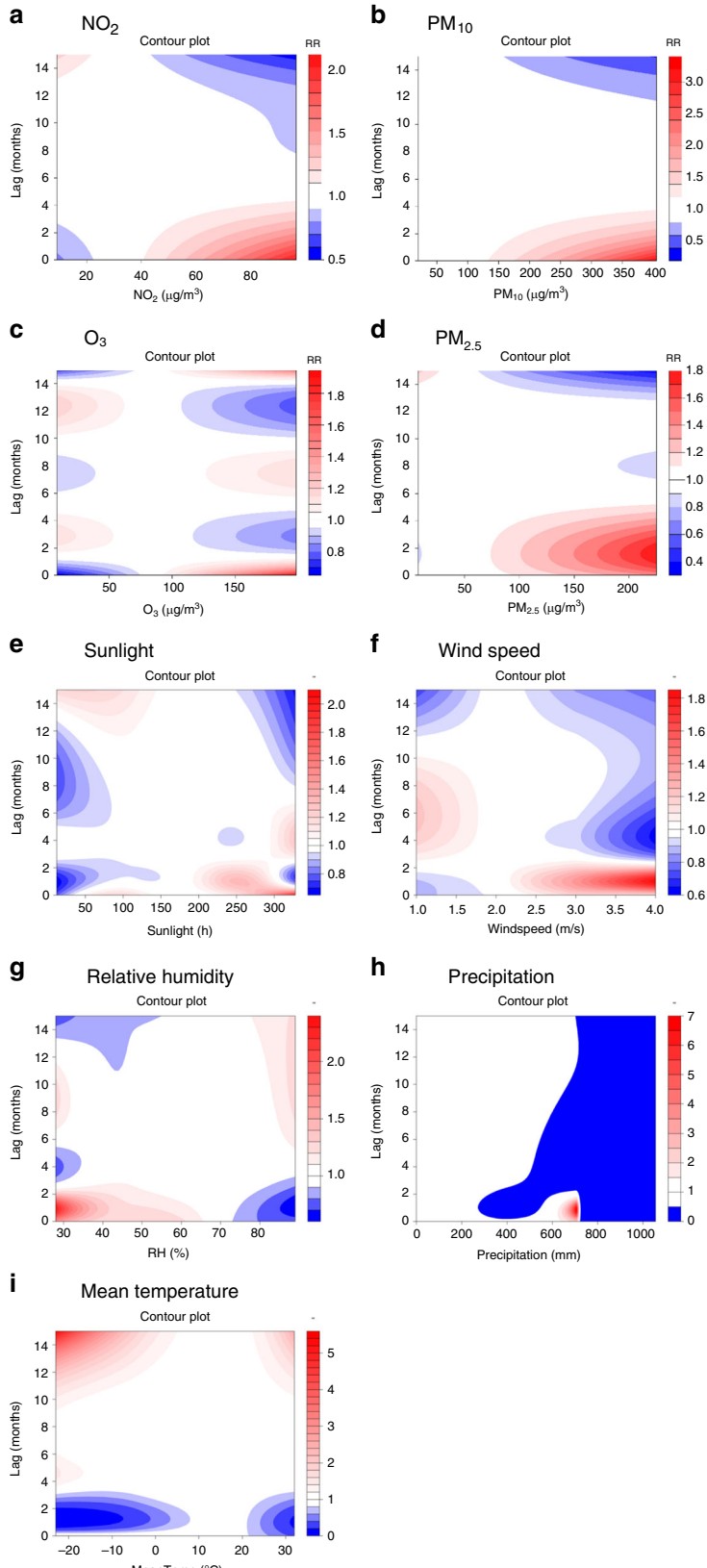

**Fig. 5 Contour plots of the exposure–response relationship for the association between scarlet fever incidence and air pollutants or meteorological conditions in the single-variable model. a–d** Four air pollutants from 2013 to 2018. **e–i** Five meteorological variables from 2004 to 2018. The reference level is set as the median value of the corresponding variable. In the single-variable model, we further adjust the temporal trend, as well as quantile groups both for average incidences and for incidence in the previous month. The $Y$-axis is the lag month ranging from 0 to 15. The $X$-axis is the range of the observed values of each variable. The color gradient represents the relative risk (RR). The red color gradient represents higher strength of RR, above 1, and the blue gradient represents lower strength of RR, below 1. The white color represents no difference, at RR = 1. Source data are provided as a Source Data file.

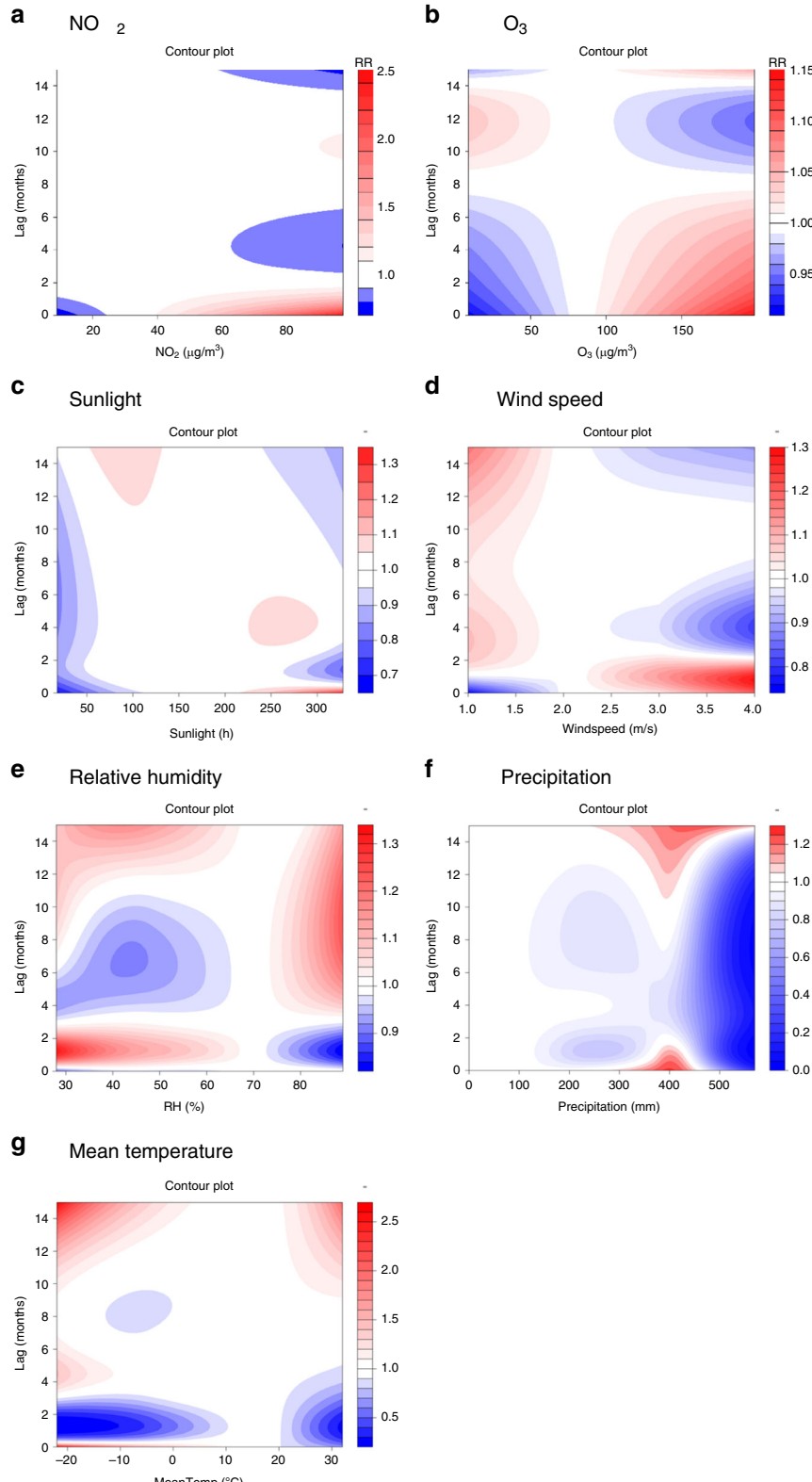

**Fig. 6 Contour plots of the exposure–response relationship for the association between scarlet fever incidence and air pollutants or meteorological conditions in the multiple-variable model from 2013 to 2018.** In addition to the adjusted air pollutants and meteorological conditions, the other adjusted variables in the model are temporal trend, the indicator variable of summer and winter breaks, and quantile groups both for average incidences and for incidence in the previous month. **a** $NO_2$ **b** $O_3$ **c** Sunlight **d** Wind speed **e** Relative humidity **f** Precipitation **g** Mean temperature. The reference level is set as the median value of the corresponding variable. The $Y$-axis is the lag month ranging from 0 to 15. The $X$-axis is the range of the observed values of each variable. The color gradient represents the relative risk (RR). The red color gradient represents higher strength of RR, above 1, and the blue gradient represents lower strength of RR, below 1. The white color represents no difference, at RR = 1. Source data are provided as a Source Data file.

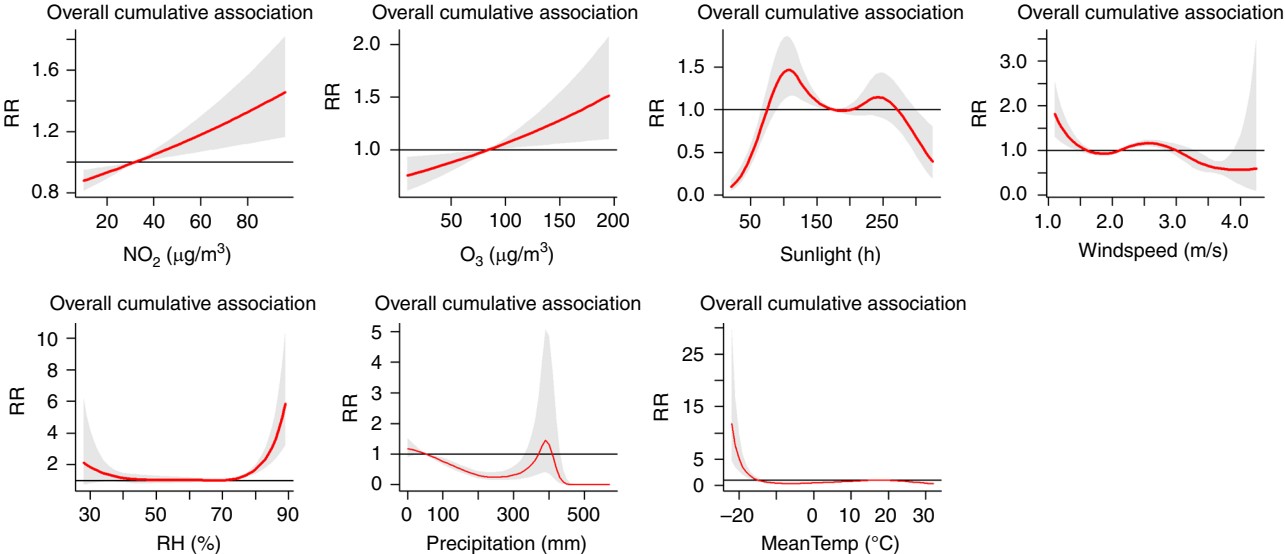

**Fig. 7 Summary of cumulative exposure–response curves of scarlet fever incidence for air pollutants ($NO_2$ and $O_3$), meteorological factors (sunlight, wind speed, relative humidity, precipitation and mean temperature) at lag 0–15 months from 2013 to 2018.** In addition to the adjusted air pollutants and meteorological conditions, the other adjusted variables in the model are temporal trend, the indicator variable of summer and winter breaks, quantile groups for average incidences, and incidence in the previous month. Source data are provided as a Source Data file.

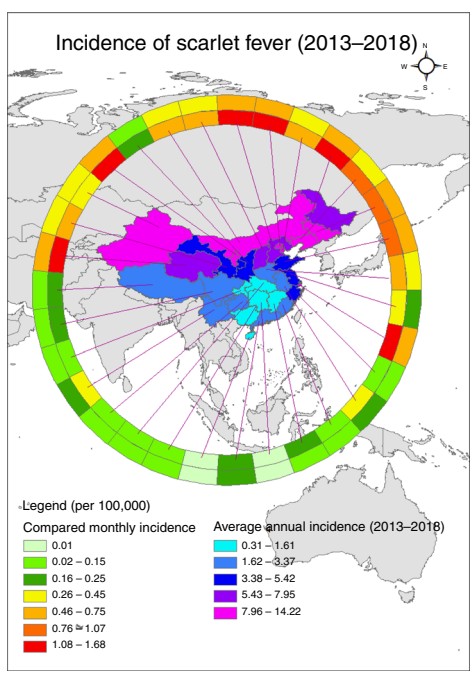

**Fig. 8 Comparison of spatiotemporal distribution of scarlet fever cases during summer and winter holidays and school terms in China, 2013–2018.** Average monthly incidence of scarlet fever per 100,000 people in the 31 Chinese provinces investigated. The outermost ring bears data for summer and winter breaks, and the innermost ring bears data for when school is in session. Choropleth maps of the average annual incidence of scarlet fever, by region, based on the annual incidence per 100,000 people in China during 2013–2018. Source data are provided as a Source Data file.

correlations with the risk of scarlet fever, and a ten-unit increment of $NO_2$ and $O_3$ concentration was associated with an increase in scarlet fever. The risks of scarlet fever are also associated with meteorological conditions (low temperature, low relative humidity, low precipitation, high wind speed, and longer sunshine).

In the past 70 years, scarlet fever incidence in China has undergone four different stages, including a high-incidence period, declining period, low incidence, and resurgence[6]. The 2011 upsurge in cases could represent a return to the upward phase in a cycle of 4–6 phases. We note that the incidence of scarlet fever increased by two times between 2011 and 2018, and the highest point in the current cycle (5.67 per 100,000 in 2018) is of greater magnitude than epidemic increases in the recent three stages' years. However, the highest incidence in China was much lower than the incidence reported in other countries and areas (e.g., 33.2 per 100,000 people in the UK[1]; 18.1 per 100,000 people in Hong Kong[5,13,20]; and 13.7 per 100,000 people in South Korea[15]). Epidemiological analysis indicates no change in the spatial and seasonal patterns or demographic features of cases during pre- and post-upsurge periods[2].

Globally, air pollution contributes substantially to disease burden, and is the fifth leading global risk factor for public health[21,22]. China's government has implemented policies and plans to reduce air pollution and its adverse effects on public health in the past[23]. The air quality of most regions has been improving since 2013, and the national annual mean $PM_{2.5}$ concentrations decreased by 77.76% (from 70.51 $\mu g/m^3$ to 39.66 $\mu g/m^3$) between 2004 and 2018. However, the high baseline levels of pollution and its subsequent health effects will persist. In our study, we found the annual levels of $PM_{2.5}$ and $PM_{10}$ were over the air quality standard (GB 3095-2012), and the concentration exceeded it in 24 of the 31 provinces. The most affected areas were in the north, in particular in high-altitude areas of China. The difference in geographical distribution of air pollutants was caused by population size and density, pollution sources, environmental policies and plans, etc.

This study investigated the possible causes of the scarlet fever upsurge, focusing on air pollutants. We found that among the six air pollutants examined, the risk estimate for $NO_2$ was the strongest. High $NO_2$ exposure is linked to higher scarlet fever incidence in high-latitude areas and during peak scarlet fever season. A novel contribution of this study is that the finding of a cumulative risk estimate for $O_3$ also had a linear exposure–response relationship with scarlet fever incidence. The risk estimate for $NO_2$ found in this study was consistent with earlier findings from Beijing[17]. $NO_2$ primarily gets in the air from the burning of fossil fuels, in

particular vehicular fuels for cars, trucks, buses, etc. in China. Ozone is primarily formed by nitrogen oxides (NOx) and volatile organic compounds (VOCs) interacting with sunlight. It is also highly related to traffic-related exhaust emissions. Scientists are trying to explain the biological mechanisms underlying the association between $NO_2$ air pollution and scarlet fever. One possible mechanism is that both short- and long-term exposure to elevated concentrations of $NO_2$ irritate airways in the human respiratory system and potentially increase susceptibility to respiratory infections[17]. Sly et al.[24] reported that the influences of $NO_2$ on the respiratory system were much higher in children, due to their immature immune systems and lungs, higher breathing rates, a greater extent of mouth breathing, and more outdoor activities. However, future patient-level and mechanistic research should be done to prove the $NO_2$ effect on the respiratory system.

Our study demonstrated effects of meteorological variables on scarlet fever during the pre- and post-upsurge periods. Specifically, in this study, it was found that the average monthly temperature showed no significant effect on scarlet fever during 2004–2010, while it became a statistically significant factor starting in 2011, contrary to a previous study in Hefei, China[25]. The contradictory results might be explained by the different study population characteristics and the meteorological factor levels. Different temperature profiles in the high- and low-temperature regions led to different temperature distributions on the thermal response curve, and finally resulted in different effects on scarlet fever. Only when mean temperatures were closer to suitable temperatures (2.8–10.6 °C in high-altitude areas and 13.6–18.1 °C in low-altitude areas) for pathogens did the incidence increase. The unexpected mechanism behind this could be that sudden temperature changes could affect the health of the respiratory epithelium at the tissue level and impair the immune system, which might add to the risk of some respiratory diseases, especially for children, who are susceptible to scarlet fever[26,27].

Our study also established that low relative humidity might have promoted the outbreaks of scarlet fever in 2011, a finding similar to a past study in England and Wales in the nineteenth century and recent studies in Hefei and Beijing of China[17,19,25]. Relative humidity might affect the ability of GAS to produce toxins and enzymes. It might also influence the spread of GAS through the vehicle of transmission. High relative humidity causes small respiratory droplets to take on water, which decreases the time of the pathogen floating in the air[18]. Relative humidity is also significantly related to airborne bacterial concentrations due to its importance to microorganism growth[28]. Furthermore, S. pyogenes may survive better and be more active in dry weather than wet conditions[17]. During the eighteenth century, scientists indicated there were positive relationships between rainfall and scarlet fever prevalence throughout Europe, Great Britain, and the US, but that observation is not consistent with our results[29]. These meteorological effects on the incidence of scarlet fever might partly account for the bacterial determinants changing from emm12 to emm1 in China during 2013–2014[30].

We found that the incidence of scarlet fever clearly decreased during the school holidays in winter (January–February) and summer (July–August) compared to when school was in session in China during 2013–2018. However, a clear upturn was observed once the new school year started. We observed a similar pattern in each subsequent year and in different provinces. Previous studies have indicated similar results. For example, Liu et al.[2] also observed that the incidence of scarlet fever decreased substantially during school holidays in China during 2004–2016; Lee et al.[31] also reported that scarlet fever incidence in Hong Kong during 2005–2015 was 32−42% lower in the week following the start of school holidays. Other childhood respiratory infections have also shown low incidence during periods of school closure[32]. This suggests that school is probably a major transmission site for scarlet fever, because children increase their social contacts substantially there, and children are most susceptible to GAS[1]. Thus, school children are the population at greatest risk, and school is the highest-risk site. In response to the prevalent risk of scarlet fever, school-based control measures could be particularly important in scarlet fever control, including not allowing students with relevant symptoms to attend school, symptom monitoring, and school closures and delayed opening in the event of outbreaks.

Several empirical studies have confirmed the ability of school closures to mitigate the spread of respiratory infectious diseases, including scarlet fever, coronavirus disease 2019 (COVID-19), influenza, and so on[33,34]. Due to the COVID-19 pandemic in 2020, we preliminarily explored the association of $NO_2$ levels with scarlet fever between January 2019 and March 2020, covering the pre-pandemic and COVID-19 pandemic period. The time series figures of incidence and $NO_2$ are listed in Supplementary Fig. 11. Compared to the past first-quarter months (January–March) from 2013 to 2019, during which the average incidence rate was 0.28 per 100,000 population, the COVID-19 epidemic period rate was 0.18 per 100,000 in Q1, 2020. The average concentration of $NO_2$ was 37.6 in Q1 from 2013 to 2019 and was 24.3 in Q1, 2020. The significant reduction in scarlet fever incidence might be attributable to several factors, including lockdown of cities with restricted movement, decreased $NO_2$ emissions, lower transmission risks and exposure risks due to self-isolation in the home, low hospital visits and consultation, and/or lower diagnosis and confirmation.

Some limitations of our study should be noted. First, we could not get air pollutants data for 2004–2012, since construction of the national air quality surveillance network did not begin until 2013. Second, social and economic status, available health services, and hygiene were not quantified precisely due to unavailability of data. Third, this study was performed based on monthly data, which means that the acute effect of meteorological and air pollutants on scarlet fever could not be investigated. Fourth, we did not analyze the effect of air pollutants and weather conditions on scarlet fever cases by GAS emm type. Fifth, this is an ecological study, and we cannot infer the effect of individual exposure levels from provinces' air pollution and meteorological conditions across large geographical areas on the risk of infection. This study can only elucidate the associations among air pollutants, meteorological conditions, and scarlet fever incidence in China by considering the temporal lagged effects. However, current evidence cannot justify any causal relationship between air pollution and scarlet fever upsurge. We further scrutinize the potential direct and indirect roles of air pollutants in facilitating the transmission of scarlet fever. Incorporating individual-level risk factors and the socio-economic environment into the risk model is warranted for future research.

In conclusion, our research shows scarlet fever cases continued to occur in China at elevated incidence rates for 8 consecutive years after a resurgence in 2011, but with decreased incidence in summer and winter school holidays each year. Based on 6 years of ecological data, we found that long-term exposure to $NO_2$ and $O_3$ is associated with scarlet fever resurgence in China, even at concentration ranges well below China's present annual mean limit. This effect was increased by low temperature, low precipitation, and relative humidity, as well as high wind speed and longer sunshine. These findings strengthen the hypothesis that air pollution is a factor in the sudden rise of scarlet fever around the world.

In the future, we should search for more evidence by further assessing the impact of $NO_2$ concentration and school closures on scarlet fever incidence because of the dramatic changes in $NO_2$ levels during the COVID-19 epidemic in China. We suggest that it is also necessary to develop school-based control measures including a school symptoms surveillance and early-warning

system, as well as air pollution surveillance in the whole of China. Finally, we suggest encouraging public health workers to consider $NO_2$ and $O_3$ risks when combatting the increasing scarlet fever trend, especially in high-latitude areas of China. A comprehensive approach is necessary to decrease the burden to society of this childhood infectious disease.

## Methods

**National Notifiable Infectious Disease Surveillance System (NNIDSS)**. After the severe acute respiratory syndrome (SARS) outbreak in 2003, the Chinese government established a real-time NNIDSS for 39 infectious diseases, covering a population of 1.3 billion in China[35]. These illnesses are divided into three classes (A, B, and C) according to the disease severity. All these diseases must be reported in a specified timeframe[35]. Scarlet fever is defined as a class B notifiable infectious disease based on criteria issued by the Ministry of Health of the People's Republic of China[2]. All probable, clinically diagnosed, and confirmed scarlet fever cases must be reported within 24 h of diagnosis online to this system[2].

**Case definitions and classification**. Scarlet fever was divided into probable, clinically diagnosed, and confirmed cases, and all these are based on diagnosis according to WS282-2008 and GB15993-1995 promulgated by the Health Ministry of China. For more details, see Supplementary Table 4.

**Long-term exposure definition**. One-year exposure to six air pollutants' concentration (particulate matter of <10 μm ($PM_{10}$) and <2.5 μm ($PM_{2.5}$), carbon monoxide (CO), nitrogen dioxide ($NO_2$), sulfur dioxide ($SO_2$), and ozone ($O_3$) was defined as an indicator of long-term exposure according to China's National Ambient Air Quality Standard (GB 3095-2012).

**Data source and collection of case data**. Four sources of data were combined to be collected throughout China. The first was obtained from the official website of the National Health Commission of the People's Republic of China (http://www.nhc.gov.cn/jkj/s3578/202004/b1519e1bc1a944fc8ec176db600f68d1.shtml) and health commissions at the province level. The second was the Chinese open access notifiable infectious disease report database (available from the Chinese Public Health Science Data Center). The third was from the Notifiable Infectious Disease Surveillance System (NNIDSS) covering the period from January 1, 2004 to December 31, 2018. Fourth, population data were from the National Bureau of Statistics of the People's Republic of China and are updated at the end of every year. The spatial distribution of average population sizes in 2018 is shown in Supplementary Fig. 12. The lowest population size was 3,370,741 persons in the Tibet Autonomous Region, and the highest was 111,689,642 persons in Guangdong Province. There were a total of 655,039 scarlet fever cases across 31 provinces during 2004–2018. We extracted data on scarlet fever, including the number of cases, incidence, and patient data stratified by onset date (month and year) and province.

**Meteorological and air pollution data source**. From January 1, 2013 to December 31, 2018, monthly air pollutant data in each province, including mean $PM_{2.5}$, $PM_{10}$, $NO_2$, $SO_2$, CO, and $O_3$, were obtained at 1498 National Air Quality Monitoring Stations in China (http://106.37.208.233:20035/) (Supplementary Fig. 13a). From January 1, 2004 to December 31, 2018, monthly meteorological data in each province were collected from the National Meteorological Information Center (http://data.cma.cn/wa). Monthly meteorological data are fully automated from 756 sites enclosed in meteorological monitoring stations in China, as shown in Supplementary Fig. 13b.

**Exposure assessment**. The basic temporal unit of meteorological conditions and air pollutants is month here. When we fit the DLNM[36], we see the relative risks at different values or concentrations at different lag months. The maximum number of lagged months is determined by finding the smallest quasi-Bayesian information criterion (QBIC) in a multivariate distributed lag nonlinear model (MVDLNM). Finally, the maximum number of lagged months is set at 15. The long-term cumulative risk here also means cumulative 15 months of risk.

**Statistical analysis**. In the descriptive analyses, mean, standard deviation, quartiles ($P_{25}$, median, $P_{75}$), minimum, and maximum were used to describe the distribution of incidence of scarlet fever, air pollutants, and meteorological variables. The comparisons of incidence before and after the epidemic surging in 2011 were conducted by z test for a Poisson distribution. The statistical level of all these analyses was set at 0.05 in two-tailed tests.

The final 12 included factors were all in monthly average form, and were $NO_2$, ozone (monthly average of daily maximum 8-h average of ozone), $PM_{2.5}$, $PM_{10}$, $SO_2$, CO, mean temperature, relative humidity, air pressure, precipitation amount, wind speed, and sunlight hours. During 2014–2018, there were some missing values for meteorological variables, including mean temperature ($n = 3$), relative

humidity ($n = 3$), air pressure ($n = 3$), precipitation ($n = 26$), wind speed ($n = 3$) and sunlight hours ($n = 63$). Thus, we used the Kalman smoothing method to impute the values by R package imputeTS (https://cran.r-project.org/web/packages/imputeTS/index.html). The descriptive statistics before and after imputing are summarized in Tables 1 and 2.

In order to reduce the confounding effects and avoid collinearity in the models, we used two approaches for analyzing the data. First, we used pairwise complete observations to compute the Pearson's correlation coefficients among scarlet fever incidence, meteorological variables, and air pollutants. We found $PM_{2.5}$ ($r = 0.8$, $p \leq 0.001$) and $PM_{10}$ ($r = 0.7$, $p \leq 0.001$) were highly correlated with $NO_2$. Thus, we did not treat $PM_{2.5}$ and $PM_{10}$ as our covariates in the MVDLNM. Second, we observed the effects from air pollutants and weather variables by single-variable and multiple-variable analysis. This comparison can help identify whether the effects are modified by other variables.

In the single-variable model, in addition to the air pollutants or meteorological conditions of interest, we further adjusted the temporal trend, quantile groups for average incidence and incidence in the previous month.

In the multiple-variable model, the variables added in MVDLNM were to control the significant confounders including $NO_2$, $O_3$, sunlight, wind speed, relative humidity, precipitation, mean temperature, temporal trend, the indicator variable of summer and winter breaks, quantile groups for average incidences and incidence in the previous month.

The overall cumulative association is composed of the sum of risks from different extents of exposure experienced within lag 0–15 months[37]. In order to compute the two cumulative effects of $NO_2$ and $O_3$ with linear exposure–response relationships, we calculated the cumulative relative risk (RR) and 95% confidence interval (CI) to express the strength of association between every 10 μg/m³ of $NO_2$ and $O_3$ and the corresponding risks of scarlet fever[38]. The reference values based on China's guidelines II of $NO_2$ and $O_3$ are 40 and 160 μg/m³, respectively.

Subgroup analyses were conducted on how selected air pollutants were associated with scarlet fever incidence by high/low population density and high-/low-incidence areas, and how selected meteorological variables were associated with scarlet fever incidence by comparing before and after the epidemic's surge in 2011. We first used DLNM[39] to compute the associations between scarlet fever incidence at lag 0 months and both selected air pollutants and meteorological variables. The predicted reference was set at the 15th percentile of concentration or values. Then, we used meta-regression[40] by R package to summarize the exposure–response relationship in each province, by predicting the 25th and 75th percentiles of population density and average incidence rate, and stratifying for time periods (before and after 2011). Finally, parameters from the meta-regression were used to predict exposure–response relationships of relative risks by subgroups.

The package "dlnm"[41] (version 2.3.9, https://cran.r-project.org/web/package/dlnm/index.html) is used to specify the cross-basis for the quadratic spline for air pollutants and meteorological variables and to predict and plot the results of a fitted model. The DLNM can help compute the relative risk of scarlet fever incidence at different levels of air pollutants or meteorological factors on different days.

First, we defined the cross-basis matrices. The cross-basis for air pollutants is specified through using the function lin, and those for the meteorological variables are specified by B-spline using the function bs from the package SPLINES in R software (version 3.6.0, https://www.rdocumentation.org/packages/splines/versions/3.6.0). Regarding the space of lags, we evaluate time lag ranging from 0 to 15 months. The maximum lag is determined by QBIC, mentioned previously. The knots for the spline for lags are placed at equally spaced values on the log scale of lags, using the function lognots. The prediction values here are centered at the median of each air pollutant and meteorological variable.

In order to visualize the exposure concentration and the incidence rates in each province, we used ArcGIS (ArcMap, version10.3; ESRI Inc., Redlands, CA, USA) to display the risk maps and also used spatial statistic Gi* to compute the spatial hotspots of the average scarlet fever incidence from 2004 to 2018. In addition, we used ring maps[42] to compare the average incidence in each province during summer (July and August) and winter breaks (January and February) and the remaining months.

**Sensitivity analysis**. For a sensitivity analysis, the choice is reduced to the identification of the optimal number and location of knots for the natural spline. Here we rely on a QBIC. The selected model has the minimum value of the sum of the QBIC in all 31 provinces. We use AIC to select the degree time variable. The model includes a natural cubic spline of elapsed time with one degree of freedom per year to control for long-term trends for air pollutants and meteorological variables in each province.

**Reporting summary**. Further information on research design is available in the Nature Research Reporting Summary linked to this article.

## Data availability

The dataset used in this study has been deposited in a publicly available website (https://doi.org/10.6084/m9.figshare.12237596.v2). The additional files required to perform this study are available from the corresponding authors upon request. Source data are provided with this paper.

## Code availability

The codes for this work are already uploaded to Figshare. It is available from the open website (https://doi.org/10.6084/m9.figshare.12237596.v2).

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

## Acknowledgements

We thank the local hospitals and prefecture Center for Disease Control and Prevention in China for collecting the data. We would like to express our sincere gratitude to Mr. Kent M. Suárez from the Research Center for Humanities and Social Science, Academia Sinica, Taipei, Taiwan for his English editing. This work was supported by the National Key Research Program (No. 2016YFC0202005), the Natural Science Foundation of China (No. 41975165) and Chongqing Science and Technology Project (No. cstc2019jscx-fxydX0035).

## Author contributions

Y. Liu, H.D., Y.B., and W.X. designed the study. H.Z., N.Z., M.W., and Y. Li performed the research. S-t.C., L.Y. and T.-C.C. performed the statistical analyses. S.L., S.Q., Y.Z., X.G., and S.W. wrote the initial paper, T.-H.L., R.L., and E.C. helped reanalyze the model, Z.Y. and S.L. contributed to subsequent revisions.

## Competing interests

The authors declare no competing interests.
