## [Peer Review File · Nature Communications]

Reviewers' comments:

Reviewer #1 (Remarks to the Author):

The authors present an interesting ecological study designed to explore the association between meteorological and air quality factors on scarlet fever incidence. The rationale behind the study was to ascertain whether changes in air pollution may underpin the rise in scarlet fever incidence in China. The authors present an array of statistical associations between the climatic factors and scarlet fever incidence, at national and provincial level. The availability of these parameters from across China over the past 6+ years are a key strength of the paper. Its main limitations are those inherent to its design, namely that of its ability to draw inferences on causality from temporal and geographical associations between observations. Thus, significant correlations may or may not reflect a causal relationship between scarlet fever incidence and air pollution. This is not acknowledged in the Discussion. Alternative hypotheses which could account for the associations observed should be considered, including behavioural factors and population density. Many methodological aspects of the paper remain unclear. Further information should be added to assist readers in understanding the relationship between air pollution (and weather) and scarlet fever incidence. In particular, the time lags fitted in their models and how 'long-term' exposure was defined. Whilst the data visualisations are attractive, they are difficult to read. What seems to be missing are more simple regression analyses showing the correlation between scarlet fever and key climatic factors, a clear description of the changing NO₂ over time and an explanation for how changes in this could lead to a sudden rise in scarlet fever.

General comments:

- The air pollution measurements were over a 6y period (or possibly 5y, according to the Discussion). As such, the title should surely reflect this.
- The abstract should outline the study design used.
- Several papers from China have already charted the rise in scarlet fever incidence; as such these 'findings' are not original to this study and should be presented accordingly.
- The meaning of 'long-term exposure' needs to be defined. Over what period of time does this refer to? And are the authors stating that they found evidence of a cumulative effect in NO₂ exposure and scarlet fever incidence?
- The description of the association between NO₂ and scarlet fever incidence and the influence of the meteorological variables would benefit from more precise description. Specifically, the use of 'triggered' is ambiguous (abstract). Are the authors stating that these meteorological factors acted as effect modifiers in some way, such that only in very specific climatic conditions did air pollutant exert an effect?
- Given the limitations of the study findings, to suggest interventions at this stage seems premature (last line of abstract and discussion).

Introduction

- Published scarlet fever surveillance data does not support that it 're-emerged' in Germany.
- Microbial and cyclical factors as drivers for the rise in scarlet fever have been explored and rejected in other studies. As such, I don't see any value in listing these.

Methods

- The population sizes covered by the provinces ought to be stated (min to max).
- The 'lag 0-15 parameterization' presumably refers to time lag – this needs to be specified clearly as to the temporal unit.

Results

- The decimal places can be reduced for ease of reading (suggest 2 dp).
- The seasonal pattern of scarlet fever and pollution levels should be briefly described

- The 'significant differences in meteorological variables' between the pre and post upsurge period should be described.

Discussion

- The limitations of the ecological study design in exploring causality need to be outlined, including inferring individual exposure levels from climatic averages across large geographical areas.
- Given the dramatic changes in NO₂ levels during the COVID-19 epidemic in China, the authors could usefully suggest further assessing the impact of this on scarlet fever incidence, albeit that the impact of school closures would also have to be factored in.

Tables & Figures

- The value of Table 1 is rather unclear given that the scarlet fever incidence and pollution parameters have changed over time.
- Figures 4 and 5 require additional information to be included in the legend to assist interpretation

Reviewer #2 (Remarks to the Author):

Include more information about scarlet fever in introduction section. Separate the Conclusion and recommendation section if it is applicable.

Responses to the reviewers' comments

Reviewer #1 (Remarks to the Author)

The authors present an interesting ecological study designed to explore the association between meteorological and air quality factors on scarlet fever incidence. The rationale behind the study was to ascertain whether changes in air pollution may underpin the rise in scarlet fever incidence in China. The authors present an array of statistical associations between the climatic factors and scarlet fever incidence, at national and provincial level. The availability of these parameters from across China over the past 6+ years are a key strength of the paper.

Response:

We are very grateful for your positive and constructive comments and suggestions, which will be very helpful to improve the manuscript. We have carefully considered your professional comments and have revised our manuscript accordingly.

Its main limitations are those inherent to its design, namely that of its ability to draw inferences on causality from temporal and geographical associations between observations. Thus, significant correlations may or may not reflect a causal relationship between scarlet fever incidence and air pollution. This is not acknowledged in the Discussion. Alternative

hypotheses which could account for the associations observed should be considered, including behavioral factors and population density.

Response:

Thank you for pointing out the misuse of this term. We have revised it accordingly. We respond to your questions as follows:

1. The main limitation of this research is inherent to its design as an ecological study. Thus, we have added these limitations in the discussion of the revised manuscript.
2. Previous studies have indicated that scarlet fever has suddenly increased in Asia and Europe in the past ten years. Scientists have tried to find the potential reasons for this re-emergence, one of which is hypothesized to be environmental variations. Some studies from Beijing, Guangzhou, and Hefei in China have already found that long-term exposure to air pollution or some weather conditions is related to the incidence of scarlet fever^[Ref 1, 2, 3], but there were limitations, including small size of some cohort studies, inconsistent results, scope being limited to local areas, and little or no attention paid to possible effects of air pollution and weather conditions.

[Ref1] Mahara G, *et al.* The association between environmental factors and scarlet Fever incidence in Beijing region: Using GIS and spatial regression models. *Int J Environ Res Public Health* **13**, (2016).

[Ref2] Lu JY, *et al.* Effect of meteorological factors on scarlet fever incidence in Guangzhou city, Southern China, 2006-2017. *Sci Total Environ* **663**, 227-235 (2019).

[Ref3] Duan Y, Yang LJ, Zhang YJ, Huang XL, Pan GX, Wang J. Effects of meteorological factors

on incidence of scarlet fever during different periods in different districts of China. *Sci Total Environ* **581-582**, 19-24 (2017).

3. Our study overcomes several limitations of previous studies, since it has the largest sample size, broad coverage across China, the longest time span, and retrospective exposure assessment stratified by area and time. Based on these, we draw inferences on the relationship between temporal and geographical associations between scarlet fever observations.

4. Also, based on your professional comments, we analyzed the associations between the scarlet fever incidence and the behavioral factors, including school closures during the summer and winter breaks, and population density in each province. We have provided new figures [Figure 8 and Supplement 15a] and updated the corresponding contents in the revised version. We hope it meets your requirement now. More details are as the following:

(1) We have conducted new analyses based on your suggestions of behavioral factors. First, we compute the two kinds of average scarlet fever incidences, either during summer (July and August) and winter breaks (January and February) or the remaining months in each province. In the following figure, the innermost ring represents the average monthly incidences during non-summer and winter breaks and the outermost ring represents the average monthly incidence in the summer and winter breaks. It clearly shows that the incidences are generally lower in the summer and winter holiday months than those non-school holidays by the

independent T-test (mean difference: 0.295 per 100 000 population, $p=0.005$). See Figure 8.

(2) We applied DLNM and meta-regression to elucidate how high and low population density is associated with the risk of scarlet fever incidence from 2013 to 2018. The predicted curves from meta-regression for the 25th (red line, population density=135.7 persons/km²) and 75th (green line, population density=480.4 persons/km²) percentiles of population showed there are no significant differences of NO₂ risks referenced at the 15 percentile of NO₂ concentration (23.32 µg/m³). See the Supplement 15a.

(3) In this revision, we considered winter and summer breaks and population size as confounders in MVDLNM (Multivariate Distributed Lag Non-Linear Model).

Figure 8. Comparison of spatiotemporal distribution of scarlet fever cases during summer and winter holidays and school terms in China, 2013-18

Notes: Average monthly incidence of scarlet fever per 100,000 people in the 31 Chinese provinces investigated. The outermost ring bears data for summer and winter breaks, and the innermost ring bears data for when school is in session. Choropleth maps of the average annual incidence of scarlet fever, by region, based on the annual incidence per 100,000 people in China during 2013-2018.

Supplement 15: Predicted exposure-response relationships in relative risk between NO₂, PM_{2.5}, and PM₁₀ (percentiles) and scarlet fever incidence in China, 2013-2018.

Notes: The predictive exposure-response relationships were modeled by two-stage methods including distributed lag non-linear model (DLNM) and multivariate meta-analysis. The reference concentrations were set at the 15th percentile of the air pollutants: 23.22 $\mu\text{g}/\text{m}^3$ for NO₂, 30.49 $\mu\text{g}/\text{m}^3$ for PM_{2.5} and 58.58 $\mu\text{g}/\text{m}^3$ for PM₁₀. The prediction values were set at the 25th and 75th percentile of population density and average incidence rate. Eight provinces were detected as hotspots and classified as high-incidence areas and the remaining provinces were classified as low-incidence areas. Those eight high-incidence provinces included Inner Mongolia, Jilin, Liaoning, Beijing, Tianjin, Hebei, Heilongjiang and Shandong. Reference is set at the 15th percentile of concentration.

Many methodological aspects of the paper remain unclear. Further information should be added to assist readers in understanding the relationship between air pollution (and weather) and scarlet fever incidence. In particular, the time lags fitted in their models and how ‘long-term’ exposure was defined.

Response:

Your comments are highly appreciated. We respond to your question following as:

1. We have adjusted as per your comments, and have provided the further information about methodological aspects, including the time lags fitted and definition of long-term exposure.

“Long term exposure definition. One-year exposure to six air pollutants’ concentration (particulate matter of less than 10 μm (PM_{10}) and less than 2.5 μm ($\text{PM}_{2.5}$), carbon monoxide(CO), nitrogen dioxide(NO_2), sulphur dioxide(SO_2), and ozone(O_3) was defined as an indicator for the long-term exposure according to China National Ambient Air Quality Standard(GB 3095-2012).”

2. The basic temporal unit is month here. When we fit the Distributed Non-linear Models (DLNM), we see the relative risks at different values or concentrations at different lag months. The maximum number of lagged months is determined by the quasi-Bayesian information criterion (QBIC). In this study, the maximum number of lagged months is selected and set to 15. In addition, we did new analysis on long-term cumulative risk, which also meant cumulative 15 months of risk.

Whilst the data visualizations are attractive, they are difficult to read.

What seems to be missing are more simple regression analyses showing the correlation between scarlet fever and key climatic factors, a clear description of the changing NO₂ over time and an explanation for how changes in this could lead to a sudden rise in scarlet fever.

Response:

Thanks for your comments on this issue. We completely agree with you.

We used simpler regression analyses showing the correlation between scarlet fever and key climatic factors, a clear description of the changing NO₂ over time, and an explanation for how changes in this could lead to a sudden rise in scarlet fever.

1. In our new Figure 4, we used pairwise complete observations to compute the Pearson's correlation coefficients among scarlet fever incidence, weather variables and air pollutants. The correlation between monthly NO₂ concentration and scarlet fever incidence in each province was 0.21 ($p \leq 0.001$). Figure 4.

Figure 4. Pearson correlation coefficients between air pollution concentrations and weather conditions and scarlet fever incidence in China, 2013-2018 (n=2232)

Notes: PM_{2.5}=particulate matter with aerodynamic diameter less than 2.5µm. PM₁₀=particulate matter with aerodynamic diameter less than 10µm. SO₂=sulfur dioxide. NO₂=nitrogen dioxide. O₃=ozone. CO=carbon monoxide. MeanTemp=mean temperature. RH=relative humidity. AP=air pressure. *means 0.05≥p>0.01; ** means 0.01≥p>0.001; *** means ≤0.001.

2. In our previous study, the provinces with a latitude higher than 33.4 degrees

north had a higher annual scarlet fever incidence than those at lower latitudes [Ref4]. To assess the potential the environment factors influence on the scarlet fever incidence in the above areas, we compared the difference in two areas. Our results supported that during 2013-2018, the mean monthly concentrations of six pollutants in the high-latitude areas (≥ 33.4 degree) is obviously different from that in the low-latitude areas (< 33.4 degree) in general, with basically higher values (Supplement 7 and 8). Nevertheless, their trends with time-series varied simultaneously.

3. The mean monthly concentrations of $PM_{2.5}$, PM_{10} and CO significantly decreased year by year, while the values of O_3 greatly increased in 2013-2018, NO_2 showed a volatile rising trend since 2016 following the downward trend during 2013-2016. The concentration of $PM_{2.5}$, PM_{10} and NO_2 in most months exceeded the China guidelines II. Meanwhile, the seasonal variation in the high-latitude areas (≥ 33.4 degree) was basically consistent with that in the low-latitude areas (Supplement 7). In addition, the concentrations of NO_2 and O_3 were positively correlated with incidences in quantile groups (Supplement 9).

Supplement 7: Time series plot of the monthly mean air pollution concentrations at high and low degrees of latitude of China, 2013-2018

Notes: PM_{2.5}=particulate matter with aerodynamic diameter less than 2.5 μm .
 PM₁₀=particulate matter with aerodynamic diameter less than 10 μm . NO₂=nitrogen dioxide.
 SO₂=sulfur dioxide. O₃=ozone. GB= National Standard of the People's Republic of China
 (Issued in 2018)

Description: The concentrations of air pollutants peaked in December to February in both high- (≥ 33.4 northern degrees) and low-latitude areas (< 33.4 northern degrees).

Supplement 8: Basic comparison of monthly air pollutants in (≥ 33.4 degrees north) and low-latitude areas (< 33.4 degrees north) of China during 2013-2018

Pollution concentration (Monthly)	Mean (high-latitude areas)	Mean (low-latitude areas)	P value
PM _{2.5} ($\mu\text{g}/\text{m}^3$)	58.6213	44.3924	<0.001
PM ₁₀ ($\mu\text{g}/\text{m}^3$)	110.8157	71.9427	<0.001
SO ₂ ($\mu\text{g}/\text{m}^3$)	31.8667	17.3056	<0.001
NO ₂ ($\mu\text{g}/\text{m}^3$)	37.0611	30.4193	<0.001
O ₃ ($\mu\text{g}/\text{m}^3$)	88.2222	83.9184	0.3473
CO (mg/m^3)	1.2255	0.9360	<0.001

Notes: In the stratified analyses, the mean concentration of five air pollutants in high-latitude regions was much higher than in low-latitude regions (all $p < 0.05$).

Supplement 9: The boxplot of six air pollutants in four quantile regions of average scarlet fever incidences from 2013 to 2018

Notes: The average incidences in 31 provinces of China are divided into four quantile regions from 2013 to 2018. The cut-off points of average incidences for the first, second, third and fourth groups are $\leq 25^{\text{th}}$ percentile, $> 25^{\text{th}}$ percentile and $\leq 50^{\text{th}}$ percentile, $> 50^{\text{th}}$ percentile and $\leq 75^{\text{th}}$ percentile, and $> 75^{\text{th}}$ percentile.

[Ref4] Liu Y, et al. Resurgence of scarlet fever in China: a 13-year population-based surveillance study. *Lancet Infect Dis* **18**, 903-912 (2018).

General comments:

- The air pollution measurements were over a 6y period (or possibly 5y, according to the Discussion). As such, the title should surely reflect this.

Response:

Thank you for your instructive comment. You have raised an excellent point. You are right. The air pollution measurements were over a six-year period (from 2013-2018). In line with your comment, we adjusted the title as “Exposure to air pollution and the resurgence of scarlet fever in

China: A nationwide six-year surveillance study”, which reflects this situation clearly. Thank you.

- The abstract should outline the study design used.

Response:

Thank you for your instructive comment, based on which we now outline the study design used; we hope it is clearer now.

The study design used in the abstract is as follows:

“In a retrospective multicenter study, we assessed 655,039 scarlet fever cases across 31 province-level administrative divisions of China. Six air pollutants’ concentration [particulate matter of less than 10 μm (PM_{10}) and less than 2.5 μm ($\text{PM}_{2.5}$), carbon monoxide (CO), nitrogen dioxide (NO_2), sulphur dioxide (SO_2), and ozone (O_3)] and six meteorological conditions were obtained monthly from National Air Quality Monitoring Stations and the National Meteorological Information Center, respectively. Ecological exposure risks to air pollutants’ concentration and meteorological conditions were evaluated by using univariate distributed lag non-linear models (DLNM) and multivariate distributed lag non-linear models (MVDLNM) and a meta-regression model. Several potential effect modifiers related to demographic and behavioral factors, including school breaks, were also examined”.

- Several papers from China have already charted the rise in scarlet fever incidence; as such these ‘findings’ are not original to this study and

should be presented accordingly.

Response:

Thank you for your valuable comment. Indeed, several papers from China have already charted the rise in scarlet fever incidence ^[Ref4,5,6]. Thus we made the descriptions related to the original findings.

[Ref4] Liu Y, *et al.* Resurgence of scarlet fever in China: a 13-year population-based surveillance study. *Lancet Infect Dis* **18**, 903-912 (2018).

[Ref5] Dong Y, *et al.* Infectious diseases in children and adolescents in China: analysis of national surveillance data from 2008 to 2017. *BMJ* **369**, m1043 (2020).

[Ref6] You Y, Davies MR, Protani M, McIntyre L, Walker MJ, Zhang J. Scarlet fever epidemic in China caused by streptococcus pyogenes Serotype M12: epidemiologic and molecular analysis. *EBioMedicine* **28**, 128-135 (2018).

- The meaning of ‘long-term exposure’ needs to be defined. Over what period of time does this refer to? And are the authors stating that they found evidence of a cumulative effect in NO₂ exposure and scarlet fever incidence?

Response: We appreciate the reviewer’s suggestion and apologize for not providing sufficient information on long-term exposure. We respond to your comments as follows:

1. We defined the long-term exposure by the data-driven approach when we set up DLNM; we used QBIC to select the maximum period of effects and the final selection is 15 months. Therefore, we evaluated the effects

of air pollutants and weather variables to the maximum 15 months lagged (the below figure is one example of NO₂ and is computed by multivariate distributed lag non-linear model (MVDLNM)). Also, the cumulative risk model meant cumulative risk of 15 months [Figure 6a].

“Long term exposure definition. One-year exposure to six air pollutants’ concentration (particulate matter of less than 10 μm (PM₁₀) and less than 2.5 μm (PM_{2.5}), carbon monoxide(CO), nitrogen dioxide(NO₂), sulphur dioxide(SO₂), and ozone(O₃) was defined as an indicator for the long-term exposure according to China National Ambient Air Quality Standard(GB 3095-2012).”

2. In this study, we did find that the cumulative effects of NO₂ and O₃ had linear exposure-response relationships after adjusting for other weather variables and air pollutants (Figure 7).

3. In addition, NO₂ and O₃ are significantly associated with scarlet fever incidence (NO₂: with reference to 40 μg/m³; O₃: with reference to 160 μg/m³) with a cumulative RR of 1.06 (95% CI: 1.02-1.10) and 1.04 (95% CI: 1.01-1.07), respectively, at a lag of 0 to 15 months by multiple variables model (Supplement 14).

4. The pooled RR of NO₂ estimated by multivariate distributed lag non-linear model varied between 0.73 and 2.44 across China (Figure 6a).

We updated the figures and the corresponding context in the revised manuscript.

Figure 6 a: Contour plots of the exposure-response relationship for the association between scarlet fever incidence and NO₂ in the multiple-variable model from 2013 to 2018

Figure 7. Summary of cumulative exposure-response curves of scarlet fever incidence for air pollutants (NO₂ and O₃), meteorological factors (sunlight, wind speed, relative humidity, precipitation and mean temperature) at lag 0-15 months from 2013 to 2018

Supplement 14: Cumulative relative risk and 95% confidence interval for the association between a 10 $\mu\text{g}/\text{m}^3$ increase in NO_2 and O_3 and scarlet fever incidence at lag 0 to 15 months in China, 2013-2018

Air pollutants	Single-pollutant model ^a	Multiple variables model ^b
NO_2	1.01 (0.97-1.04)	1.06 (1.02-1.10)
O_3	1.04 (1.01-1.07)	1.04 (1.01-1.07)

- The description of the association between NO_2 and scarlet fever incidence and the influence of the meteorological variables would benefit from more precise description. Specifically, the use of ‘triggered’ is ambiguous (abstract). Are the authors stating that these meteorological factors acted as effect modifiers in some way, such that only in very specific climatic conditions did air pollutant exert an effect?

Response:

Thank you for your comment. We have now described the association between NO_2 and scarlet fever incidence and the influence of the meteorological variables more clearly. We did find long-term exposure to ambient NO_2 was associated with scarlet fever incidence, and also found that high O_3 , low temperature, and low wind speed were associated with the increase of scarlet fever incidence (Figure7). The non-linear associations were found among sunlight hours, relative humidity and

precipitation.

1. In order to reduce the confounding effects, we used two approaches. First, we computed the correlation coefficients to evaluate the collinearity among incidence, weather variables and air pollutants. We found $PM_{2.5}(r=0.8)$ and $PM_{10}(r=0.7)$ were highly correlated with NO_2 . Thus, we did not treat $PM_{2.5}$ and PM_{10} as our covariates in the multivariate distributed lag non-linear model (MVDLNM).
2. Second, we observed the effects from air pollutants or weather variables by univariate and multivariate analysis. This comparison can help identify whether the effects are modified by other variables. We reanalyzed the original Figure 4 and Figure 5 by univariate DLNM with consideration of the lag time in each province and adjusted the degrees of freedom from 5 to 1 due to the loss of data in the first 15 lagged months.
3. In addition, we added new analyses from MVDLNM to control the confounders including NO_2 , O_3 , sunlight, wind speed, relative humidity, precipitation, mean temperature, temporal trend, the indicator variable of summer and winter breaks, quantile groups for average incidences and incidence in the previous month.

We have now described the association between NO_2 and scarlet fever incidence and the influence of the meteorological variables more clearly. We hope it meets your requirement now.

- Given the limitations of the study findings, to suggest interventions at this stage seems premature (last line of abstract and discussion).

Response:

Thank you; you have raised an excellent point. We completely accept your suggestions. We have deleted the interventions (decrease the NO₂ emissions) in the last line of the abstract and discussion. We have replaced it with the following sentence in the abstract and we have also deleted the last paragraph of the discussion.

“This study should encourage public health authorities to consider NO_x and O₃ risks when addressing the prevention and control of scarlet fever resurgence. And school-based control measures could be particularly important in scarlet fever control”.

Introduction

- Published scarlet fever surveillance data does not support that it ‘re-emerged’ in Germany.

Response:

Thank you for your instructive comment. You are right. Published scarlet fever surveillance data ^[Ref7] did not support the conclusion that it ‘re-emerged’ in Germany. They only supported the conclusion that the incidence of scarlet fever among German children younger than 10 years (280-550 cases per 100 000 children) has constantly been high. We accept

your opinion, and have made the correction.

[Ref7] Brockmann SO, Eichner L, Eichner M. Constantly high incidence of scarlet fever in Germany. *Lancet Infect Dis* **18**, 499-500 (2018).

- Microbial and cyclical factors as drivers for the rise in scarlet fever have been explored and rejected in other studies. As such, I don't see any value in listing these.

Response:

We appreciate the reviewer's comments. We have fully accepted your opinion and deleted this sentence.

Methods

- The population sizes covered by the provinces ought to be stated (min to max).

Response:

Thank you. We have provided the spatial distribution of average population sizes in 2018 (min to max) covered by Supplement 2. The lowest population size was 3,370,741 persons in Tibet and the highest population size was 111,689,642 persons in Guangdong Province of China.

Supplement 2: The spatial distribution of average population in 2018 in 31 provinces of China

Notes: Choropleth maps of the average population, by region, in China in 2018.

- The ‘lag 0-15 parameterization’ presumably refers to time lag – this needs to be specified clearly as to the temporal unit.

Response:

We defined the long-term exposure by the data-driven approach when we set up DLNM; we used QBIC to select the maximum period of effects, and the final selection is 15 months. Therefore, we evaluated the effects of air pollutants and weather variables to the maximum 15 months lagged. We made the correction for this part of the method. Thank you.

Results

- The decimal places can be reduced for ease of reading (suggest 2 dp).

Response:

Thank you for your comment. We fully agree with using 2 dp.

We have revised the manuscript, including Figure 1, accordingly.

- The seasonal pattern of scarlet fever and pollution levels should be briefly described

Response:

Thank you very much for your useful comment. We clearly described the seasonal pattern of scarlet fever and pollution levels in the results. Please refer to the revised manuscript. We used box-plots to display the seasonal pattern of scarlet fever, weather conditions and pollution levels.

1. The seasonal pattern of air pollutants: Most air pollutants had higher concentration in the winter and spring, but O₃ was higher in summer and spring. See Figure 2a.
2. The seasonal pattern of weather condition: The temperature, relative humidity and precipitation were higher in summer. The atmosphere pressure was higher in winter, and wind speed and sunlight were higher in spring. See Figure 2b.
3. The seasonal pattern of scarlet fever: The average incidence of scarlet fever was highest in spring. See Figure 2c.

a. Air pollutants

b. Meteorological conditions

c. Number of scarlet fever cases

Figure 2. Boxplots of six air pollutants, six meteorological conditions and the number of scarlet fever cases in four seasons from 2013 to 2018

- The ‘significant differences in meteorological variables’ between the pre and post upsurge period should be described.

Response:

Your comment is helpful; thank you. We used box-plots and independent

T tests to display the differences of meteorological variables between the pre- (before 2011) and post-upsurge (after 2011) periods. Among six variables, we found there were three variables reaching statistical significance ($p < 0.05$), including precipitation, wind speed, and sunlight hours. See Supplement 10. In addition, we computed the predicted exposure-response relationships in relative risk for those three meteorological variables referenced at the 15th percentile (Supplement 16). We cannot find significant differences before and after the surge.

Supplement 10: Box-plots of six weather variables before and after 2011 in China

Notes: There are three significantly different variables ($p < 0.05$) including precipitation, wind speed, and sunlight hours. RH= Relative humidity; Ap=Air pressure; WS=Wind speed.

Supplement 16: Predicted exposure-response relationships in relative risk between monthly wind speed, precipitation, sunlight (percentiles) and scarlet fever incidence before and after 2011

- a. Precipitation
- b. Wind Speed
- c. Sunlight

Notes:

- a. Excludes three provinces including Tianjin and Beijing because of extreme outliers and Hainan because of many zero cases in months
- b. Excludes Hainan because of many zero cases in months
- c. Excludes Hainan because of many zero cases in months

Reference is set at the 15th percentile.

Discussion

- The limitations of the ecological study design in exploring causality need to be outlined, including inferring individual exposure levels from climatic averages across large geographical areas.

Response:

Thank you for your instructive comment. We outline the limitations of the ecological study design in exploring causality, including inferring individual exposure levels from climatic averages across large geographical areas.

- Given the dramatic changes in NO2 levels during the COVID-19 epidemic in China, the authors could usefully suggest further assessing the impact of this on scarlet fever incidence, albeit that the impact of school closures would also have to be factored in.

Response:

Thank you for your comment. You raised a very good question. You are right. The NO₂ levels have dramatic changes during the COVID-19 epidemic in China because some cities have been locked down, most people have had to stay at home and all schools are closed and delayed open.

1. Based on your comment, we further analyzed the impact of NO₂ levels on scarlet fever between January 2019 and March 2020, covering the pre-pandemic and COVID-19 pandemic period. The time series figures of scarlet fever incidence and NO₂ are listed below. Compared to the past first-quarter months (January to March) from 2013 to 2019, during which the average scarlet fever incidence rate was 0.28 per 100 000 population, the COVID-19 epidemic period rate was 0.18 per 100 000 in Q1, 2020. However, the average concentration of NO₂ was 37.6 in Q1 from 2013 to 2019 and was 24.3 in Q1, 2020. See Supplement 17.

2. Also, we added the impact of school closures on the scarlet fever incidence (Figure 8). During the COVID pandemic, lower scarlet fever incidence was reported due to several factors, including lockdown of cities with restricted movement, lower transmission risks due to self-isolation in the home, low hospital visits and consultation, and lower diagnosis and confirmation.

3. In the Discussion section, we added your opinions in the last paragraph.

This is a most important issue in future research.

Supplement 17: Time series plot of the monthly mean NO₂ pollution concentrations and scarlet fever incidence in the overall population in China, January of 2013 to March of 2020

Tables & Figures

- The value of Table 1 is rather unclear given that the scarlet fever incidence and pollution parameters have changed over time.

Response:

Thank you for this comment. In the table, it is hard to display the seasonal pattern of the incidence and air pollutants. We used box-plots to reflect the temporal changes, which are shown in the previous replies.

- Figures 4 and 5 require additional information to be included in the legend to assist interpretation

Response:

Thank you for your comment. Your comments are very helpful to improve our manuscript. We improved Figures 4 and 5 and provided additional information in the figure legends for these two figures. We think it is clear now.

Reviewer #2 (Remarks to the Author)

Include more information about scarlet fever in introduction section.

Separate the Conclusion and recommendation section if it is applicable.

Response:

Thank you for your instructive comment. Actually, you raised very good questions. All your comments are very helpful to improve our manuscript. Following your comments, we have added more disease information about scarlet fever in the Introduction section based on the published articles. Also, we separated the Conclusion and Recommendation sections in the Discussion. Thank you very much.

REVIEWERS' COMMENTS:

Reviewer #1 (Remarks to the Author):

The authors have adequately addressed the points by my original review with the exception of one request. Whilst the revised manuscript now highlights the limitation of an ecological study design in terms of not being able to infer exposure at an individual /person level, they have missed a more fundamental point, namely that association does not equate to causation. This should be acknowledged within the Discussion.

Other minor points:

Page 3 – the 'etc' after United Kingdom should be removed. To the best of my knowledge, no other countries in Europe have reported a sudden increase in SF.

Refs – suggest these are checked as I spotted an error in the author name for ref 1 (first name rather than surname given).

Responses to the reviewers' comments

Reviewer #1 (Remarks to the Author)

The authors have adequately addressed the points by my original review with the exception of one request. Whilst the revised manuscript now highlights the limitation of an ecological study design in terms of not being able to infer exposure at an individual /person level, they have missed a more fundamental point, namely that association does not equate to causation. This should be acknowledged within the Discussion.

Response: The reviewer has raised an excellent point. We agree with you that association does not equate to causation. We have acknowledged this limitation in the discussion. Thank you for your instructive comment.

Other minor points:

Page 3 – the 'etc' after United Kingdom should be removed. To the best of my knowledge, no other countries in Europe have reported a sudden increase in SF.

Response: Thanks for pointing out this mistake. We have made this correction.

Refs – suggest these are checked as I spotted an error in the author name for ref 1 (first name rather than surname given).

Response: Thank you very much for your instructive comment. All your professional comments will be helpful to improve our manuscript. We have checked all references to ensure they conform to the Nature Communications style.